# Acute Myelopathy in Childhood

**DOI:** 10.3390/children8111055

**Published:** 2021-11-15

**Authors:** Giulia Bravar, Aphra Luchesa Smith, Ata Siddiqui, Ming Lim

**Affiliations:** 1Department of Paediatrics, Hospital Santa Maria della Misericordia, 33100 Udine, Italy; giuliabravar@gmail.com; 2Whittington Hospital, London N19 5NF, UK; aphra.luchesasmith1@nhs.net; 3Children’s Neurosciences, Evelina London Children’s Hospital, Guy’s and St. Thomas’ NHS Foundation Trust, London SE1 7EH, UK; ata.siddiqui@gstt.nhs.uk; 4Department of Neuroradiology, King’s College Hospital, London SE5 9RS, UK; 5Department of Women and Children’s Health, School of Life Course Sciences, Faculty of Life Sciences and Medicine, King’s College London, London SE5 9NU, UK

**Keywords:** myelopathy, myelitis, demyelination, spinal cord injury

## Abstract

Acute myelopathy presenting in childhood can be clinically classified based on the location of injury (with resulting spinal syndrome) or the cause (broadly traumatic or non-traumatic). Types of nontraumatic myelopathy include ischaemic, infectious, inflammatory, nutritional, and metabolic causes, some of which may be part of a systemic illness such as systemic lupus erythematosus or a demyelinating disease such as multiple sclerosis. Nonaccidental injury is an important consideration in cases of traumatic myelopathy, which may often be associated with other injuries. Assessment should include neuroimaging of the brain and spinal cord, with further investigations targeted based on the most likely differential diagnoses; for example, a child with suspected demyelinating disease may require specialist cerebrospinal fluid and serological testing. Management also will differ based on the cause of the myelopathy, with several of these treatments more efficacious with earlier initiation, necessitating prompt recognition, diagnosis, and treatment of children presenting with symptoms of a myelopathy. Important components of holistic care may include physiotherapy and occupational therapy, with multidisciplinary team involvement as required (for example psychological support or specialist bowel and bladder teams).

## 1. Introduction

Myelopathy refers to any disorder leading to neurologic deficits originating from the spinal cord. Injury sustained is generally categorized as traumatic or nontraumatic in etiology. A recent literature review showed a global median incidence for traumatic spinal cord injury ranging from 3.3/million/population/year in Western Europe to 13.2/million/population/year in North America, with land transport accidents being the leading cause [1]. Traumatic spinal cord injury often leads to death or lifelong disability, with a 30% increased annual mortality rate in under 16s compared with that of older patients [1].

Nontraumatic myelopathies can result from inflammatory, compressive, infectious, or vascular disorders. The recently reported global incidence of nontraumatic spinal cord injury varies between 2.1/million/population/year in North America to 6.5/million/population/year in Australasia [1]. A single-centre retrospective study conducted in Ireland showed acute transverse myelitis (ATM) to be the most common cause of nontraumatic acquired myelopathy, followed by vascular disorders [2]. ATM is an immune-mediated central nervous system (CNS) disorder which may be idiopathic or related to an underlying demyelinating, infectious, or autoimmune disease [3]. ATM carries a high risk of long-term disability, with early disease recognition and treatment important in the management of these patients [3]. This review will focus on acquired traumatic and nontraumatic myelopathies, encompassing inflammatory, infectious, and vascular causes, with an overview of clinical syndromes, diagnosis, and outcomes of these conditions in childhood. Chronic and progressive types of myelopathies (nutritional, genetic, and metabolic disorders) will be overviewed briefly. We also provide a diagnostic algorithm directed at the assessment and characterisation of paediatric myelopathies.

## 2. Clinical Syndromes

In the evaluation of a myelopathy, different clinical syndromes can be distinguished reflecting the neuroanatomical localization of the lesion and the spinal tract or tracts involved, as detailed in Table 1. Lesions may be complete (or transverse), manifesting with bilateral sensory loss, motor deficits, and dysautonomia. The level of the lesion will determine different clinical syndromes: upper cervical lesions result in spastic tetraplegia (upper motor neuron involvement), incontinence and sensory loss below the level of the lesion whilst lower cervical lesions cause weakness, areflexia, and fasciculation in the upper limbs (lower motor neuron lesion) with spastic paraparesis, incontinence, and sensory loss below the lesion [4].

A partial cord syndrome instead suggests involvement of a specific cord tract, for example injury to the lateral half of the spinal cord (e.g., spinal cord trauma or compression) resulting in the Brown–Sequard syndrome [5]. A selective tract involvement (e.g., selective posterior column) is usually associated with metabolic or degenerative chronic myelopathies (such as Vitamin B12 deficit) [5]. Isolated motor involvement characterized by flaccid paralysis, fasciculation and areflexia is instead suggestive of prominent injury to anterior horn cells (lower motor neurons), such as in acute flaccid myelitis (AFM) caused by poliovirus, other enteroviruses, or flaviviruses [6].

The spinal cord also plays a central role in autonomic nervous system (ANS) pathways, hence lesions give rise to variable dysautonomic manifestations, depending on the level and severity of injury [7]. In general, lesions above T1 result in complete loss of sympathetic outflow resulting in bradycardia, arrhythmias, hypotension, and loss of thermoregulation, acutely described as neurogenic shock. For injuries below T4, cardiac responses will be maintained, with central regulation on vascular tone and blood pressure lost. Lesions above T6 result in impairment of the splanchnic vasomotor response; this mechanism is thought to be implicated in the occurrence of autonomic dysreflexia [7,8]. Moreover, patients with spinal cord injuries may suffer from variable degrees of bowel, bladder, and sexual dysfunction; these may be the main manifestation of injury, as in cauda equina syndrome [9].

Beyond distinctions made on the basis of injury localization, clinical evolution over time should be also considered. Any acute spinal cord injury involving the corticospinal tracts manifests initially with areflexia and flaccid paralysis, and then evolves into spasticity and hyperreflexia below the level of the lesion [10]. Acute autonomic impairment (defined by cardiovascular dysfunction, atonic bladder, and paralytic ileus) [7] also evolves, depending on injury type and localization, into a chronic condition marked by various types of voiding and bowel dysfunction (i.e., hypertonic bladder and faecal incontinence), impairment in vascular tone regulation (such as autonomic dysreflexia or orthostatic hypotension) and temperature homeostasis [7,10].

Timing of the onset of symptoms and their progression may also be of use in determining the aetiology of acute myelopathies. A hyper-acute presentation suggests a vascular cause, whilst a subacute onset of symptoms, progressing to a nadir within days to weeks, is more indicative of an inflammatory myelopathy such as ATM [3].

## 3. Causes of Childhood Myelopathy

### 3.1. Traumatic

Traumatic spinal cord injury (TSCI) is rare in younger children, with 90% occurring in patients aged 15 or older [14]. Biomechanical characteristics of the immature spine, a larger head to body ratio, and diverse mechanisms of trauma determine distinct patterns of injury in the paediatric age group. In contrast to adults, the cervical spine is involved in 60–80% of injuries, while the thoracolumbar spine is affected in only 5 to 34% of cases [14], more commonly in older children [15].

Below the age of eight, greater laxity of ligaments and the presence of shallow and horizontal facet joints, underdeveloped uncinate processes, and weak nuchal muscles allow greater mobility of the spine, especially in the cervical area [16,17]. The larger head to body ratio results in a different fulcrum of motion of the cervical spine (located at C2–C3) and a subsequently higher level of injuries in this age group [17,18]. These factors explain some of the physiological findings typical of younger children (i.e., greater atlanto-dens interval, pseudo-subluxation of C2 on C3 [19], as well as their higher vulnerability to cervical cord injuries [17].

A child’s age also correlates with injury dynamics. Motor vehicle accidents are the primary cause of spinal injury in the younger child, whilst sports injuries are more common in adolescents [18,20]. Spinal cord injury at birth is rare and most often related to a traumatic delivery, although it was reported also after uncomplicated vaginal delivery with the high cervical spinal cord most commonly involved [21,22]. The possibility of non-accidental injury should always be considered when evaluating a child with spinal cord trauma, especially under two years of age [23]. In this context, spinal injury is commonly associated with other injuries, often multiple skeletal fractures and head trauma, and is unrecognized at clinical presentation in up to 50% of cases [24].

The increased flexibility of the immature spine also explains the higher incidence of spinal cord injury without radiographic abnormality (SCIWORA) in children. Initially recognized as a distinct clinical entity, SCIWORA is characterized by the presence of neurological dysfunction consistent with spinal cord injury in the absence of radiological findings on X-ray or CT [25]. With the increasing use of magnetic resonance imaging (MRI), however, evidence of spinal nonbony damage and cord injury was demonstrated in the majority of SCIWORA cases [26,27]. The exact prevalence of SCIWORA is uncertain; it was reported in 33% of spinal cord trauma, most commonly in children younger than eight and predominantly involving the cervical spine [25,28]. Delayed onset of symptoms to up to 4 days after trauma was reported in 52% of cases [25]. Younger age, evidence of cord injury on MRI, and severity of neurological symptoms may predict poorer outcomes [26,27]. Figure 1A shows two examples of MRI imaging in paediatric traumatic spinal cord injury. Reviews further exploring different types of traumatic spinal cord injuries cover detail beyond the scope of this article [16,19,29].

### 3.2. Ischaemia

Spinal cord ischemia is a rare event increasingly recognized in children, typically characterized by rapid progression of sensorimotor symptoms within minutes to hours [30]. Depending on the spinal cord artery involved, spinal cord ischemia may present as an anterior or posterior spinal cord syndrome [12]. Diffuse cord ischemia may result from global hypotension with secondary hypoperfusion of the spinal cord (for example post cardiac arrest, cardiac or aortic surgery). Focal spinal infarcts are rare in children and frequently difficult to differentiate from acute inflammatory transverse myelitis. The thoracolumbar region and anterior two-thirds of the spinal cord are more vulnerable to ischemia due to their lesser vascular supply, hence the anterior cord syndrome is the most common clinical presentation [11,31].

MRI may aid diagnosis, showing an hyperintense spinal cord on T2 weighted images with a “pencil-like” shape [32]. Figure 1B shows three examples of paediatric spinal cord infarcts on MRI. Diffusion-weighted imaging may be useful to confirm diagnosis during the first 7 to 10 days [33]. Spinal cord ischemia may also occur during spinal injuries leading to arterial disruption or thrombosis, and was reported even in the context of minor trauma [30]. Fibrocartilaginous emboli or reactive vasospasm of spinal cord arteries were suggested as possible causative factors [26,34], with other described etiologies of spinal cord ischemia including iatrogenic intravascular injury, cerebellar herniation, thrombotic or embolic disorder, inflammatory vascular disorders, atlantoaxial instability, and spinal arteriovenous malformations [30]. In children, the cause of spinal cord infarction often remains unclear or is found to be multifactorial [30]. Treatment is directed at the underlying etiology (for example anticoagulation for thrombotic disorders, steroids for vasculitis) [30]. Outcome, reported mainly in adults, is generally poor [32,35]. Table 2A contains a brief summary of vascular myelopathies.

### 3.3. Inflammatory

One inflammatory spinal cord disorder that is an important cause of non-compressive myelopathy is acute transverse myelitis (ATM) [36]. Paediatric patients account for up to 20% of all ATM cases, with a reported incidence of around 2/million children/year, highest in those aged younger than 5 and older than 10 years [37,38]. The term transverse myelitis refers to the clinical picture characterized by “transverse” sensory deficits rather than the spinal cord pathology [39]. Onset is often preceded by infection [3,37,40]. The Transverse Myelitis Consortium Working Group (TMCWG) provided diagnostic criteria for acute idiopathic transverse myelitis, which outline the distinction between idiopathic ATM and disease-associated ATM [36].

ATM presents with acute onset motor, sensory, and autonomic symptoms. Back pain is often the first described symptom followed by motor and sensory deficits, which may be asymmetric. Sensory symptoms can be either positive (burning, hyperaesthesia, allodynia) or negative (numbness) and a clear sensory level may not be detectable, especially in younger children. Sphincter dysfunction is common during the acute phase [37,38,40]. Spinal shock characterized by areflexia and reduced sympathetic activity below the level of injury may be evident in the acute phase, with a subsequent risk of developing autonomic dysreflexia [39]. In infants, the clinical presentation may be subtle, with irritability, reduced movement and decreased urinary output; priapism was also described in children [41].

The American Spinal Injury Association (ASIA) impairment scale should be used to evaluate the extent and the location of injury [42], although it may be difficult to apply to very young children [43]. Disease progression usually reaches a nadir in the first week [37,38] and always within 21 days [36]. A plateau then follows, before a recovery phase begins [37,38,40]. Pain is usually the first symptom to resolve, followed by motor deficits. Sensory symptoms and bladder function tend to recover last [40].

Within disease-associated ATM, the three main categories are infectious myelitis (e.g., herpes viruses), systemic inflammatory disorders (e.g., systemic lupus erythematosus) and acquired demyelinating disorders, where ATM may represent a first presentation (multiple sclerosis (MS) [44], acute disseminated encephalomyelitis (ADEM) [45], neuromyelitis optica spectrum disorders (NMOSD) [3] and the increasingly recognised myelin oligodendrocyte glycoprotein antibody associated disease (MOGAD) [46]). The presence of concomitant encephalopathy or polyfocal neurological deficits is not typical of isolated ATM and should raise suspicion of other conditions such as ADEM or MOGAD [3,45,46].

Systemic inflammatory disorders should be investigated for detailed history and examination, looking for multisystem involvement (such as rashes, arthritis, uveitis, or nephropathy). Laboratory tests including full blood count, inflammatory markers and autoantibodies should be included in the initial diagnostic workup [3]. In up to 50% of NMOSD cases, ATM represents the first clinical event, either in isolation or in association with optic neuritis; therefore, testing for AQP4-antibodies should be considered in all patients [3,47].

MRI of the spine is key to the diagnosis of paediatric ATM, usually revealing central cord hyperintensities on T2 weighted images involving both grey and adjacent white matter; T1 hypo-intensities may be present in one-third of cases. Lesions are usually located in cervicothoracic regions [38,48] and may extend over three or more segments; this is described as a longitudinal extensive transverse myelitis (LETM). LETM was reported in up to 85% of ATM in children [3,42,48] and represents a core clinical characteristic in NMOSD [47], although it may also be seen in MS [37]. MRI of the brain must be included in initial investigations, as asymptomatic brain lesions were reported in 40% of children with idiopathic ATM and represent a risk factor for subsequent relapse [42]. Figure 2 shows classic examples of MRI in four demyelinating diseases that can present with ATM. Table 2B summarises key characteristics of inflammatory myelopathies.

In some cases, spinal injury may be absent in the initial scan; however, a repeat scan in 5–7 days may reveal subsequent spinal cord atrophy [38,48]. Cerebrospinal fluid (CSF) analysis shows pleocytosis and increased protein levels in approximately 50% of children, with oligoclonal bands detected in 1/3 of cases [3]. Serological investigations are important to exclude ongoing autoimmune disease and infection. One important mimic of ATM is Guillain–Barre syndrome (GBS). However, GBS is characterized by bilateral rapidly progressive symptoms in the absence of spinal cord involvement on MRI; these factors may help to distinguish the two conditions [39].

The first line of treatment for ATM consists of high dose steroids. In children, methylprednisolone is the medication of choice, at the dosage of 30 mg/kg/dose once a day for 3–5 days [49]. In disease-associated ATM, targeted therapy should be considered (e.g., cyclophosphamide for SLE or plasma exchange for NMOSD) [50]. Plasma exchange has proven benefits as second line therapy [49,51], whilst the evidence for using intravenous immunoglobulin is inconclusive [49,52].

Approximately 50% of paediatric patients will experience complete recovery [38], with overall better outcomes than adults. However, incomplete recovery may mean lifelong severe disability [37]; a proportion of this may be preventable through early effective diagnosis and treatment [51]. Patients may also progress to relapsing disease in the context of ATM, MS, NMOSD or underlying systemic autoimmune disease [42].

### 3.4. Infectious

Viral and bacterial infections can affect the spinal cord and result in acute or chronic myelopathy. Bacterial infections, such as spinal epidural abscesses or spinal cord abscesses, are infrequent in the paediatric age group, with the intramedullary abscess an even rarer disorder with approximately 100 cases reported in the literature. In children, spread of infection from the skin is common, with almost half of patients having an underlying anatomical defect, most commonly a dermal sinus. Clinical presentation may be acute, subacute or chronic [53], characterised by evidence of infection (i.e., fever, leucocytosis) and spinal cord compression (pain and neurological deficits) [53], which in chronic cases may mimic a spinal tumour [54]. Recurrent bacterial meningitis has also been described as an atypical presentation of intramedullary abscess [55]. MRI with gadolinium enhancement is key to diagnosis, with CSF analysis often unremarkable [53]. Prompt surgical drainage and antibiotics (initially broad spectrum to cover gram positive, negative and anaerobic bacteria) is the treatment of choice [53]. Epidural spinal abscesses can present with a similar clinical picture. Epidemiology, pathogenesis, and management in children was recently reviewed elsewhere [56,57]. Other forms of bacterial nonpyogenic infections of the spinal cord reported in children (such as Lyme disease) may instead present with transverse myelitis and should be considered in the differential diagnosis of ATM [58,59,60]. Figure 1(C2) shows an example of tuberculous meningitis with ring enhancing lesions in the spinal canal with associated extensive myelopathy.

Viral infections may cause acute myelopathy as a consequence of the host’s response to infection or through direct cytolytic viral damage of neuronal cells [6]. The former mechanism is thought to be involved in those acute viral myelopathies that present with a clinical picture of ATM due to herpesviruses (such as CMV (shown in Figure 1(C1)) HHV6 and 7, HSV 1 and 2, VZV and EBV). These cases are characterised by diffuse involvement of spinal cord white and grey matter, manifesting as sensory, motor, and autonomic deficits [6].

The later mechanism of direct cytolytic damage may be involved in acute flaccid myelitis (AFM). AFM refers to a specific form of myelitis resulting in acute onset flaccid paralysis with relative sparing of sensory and autonomic functions, historically caused by polioviruses. Acute poliomyelitis (polio) is characterised by rapidly progressive asymmetrical weakness, affecting predominantly the lower limbs without any objective sensory loss, due to the prominent involvement of the anterior horn cells of the spinal cord [61]. With the ongoing eradication of Polioviruses, other viruses were found to cause Polio-like disease, such as Enteroviruses (i.e., Enterovirus 71, 68; Coxsackie viruses) and Flaviviruses (i.e., West-Nile Viruses, Japanese Encephalitis Virus) [6]. The term Acute Flaccid Myelitis was adopted by the Centre for Disease Control (CDC) since 2015 to define acute onset flaccid limb paralysis with evidence of spinal cord lesions restricted to grey matter on MRI, with or without cerebrospinal fluid pleocytosis.

Enterovirus 71 is one of the main causative agents of hand, foot and mouth disease but may occasionally affect the CNS, mainly through aseptic meningitis, rhomboencephalitis or a polio-like AFM [62,63]. Enterovirus D68 (EV D68) recently emerged as another potential cause of AFM. During a large outbreak of EV-D68 respiratory infection in 2014 (Colorado, USA), 120 paediatric cases of AFM were reported, however a clear causal link with EV-D68 was not demonstrated [64,65]. Figure 1D shows the imaging of a patient with AFM possibly precipitated by EV-68.

West Nile Virus (WNV) is a Flavivirus transmitted primarily by mosquitoes. WNV infection can either be asymptomatic, cause a flu-like syndrome (fever, rash, muscle pain, headache), or manifest as West Nile Neuroinvasive Disease (WNND) with meningitis, encephalitis or acute myelopathy [66]. Children account for 4% of all WNND cases. Spinal cord involvement is less common than in adults [67], and may present with an acute flaccid paralysis with evidence of anterior spinal cord involvement on MRI [68]. Diagnosis is based on the identification of specific IgM antibodies in serum or CSF, or the detection of WNV nucleic acid by amplification assays [69]. The outcome of WNV-associated AFM is poor with persistent motor weakness common, and most recoveries seen in the first 6–8 months from disease onset [70].

Viruses such as HTLV-1 and HIV may also be involved in the pathogenesis of progressive and chronic myelopathies [71,72]. Only rarely were these conditions reported in children. A brief summary of infectious myelopathies can be found in Table 2A.

### 3.5. Neurodegenerative, Metabolic, and Nutritional

Nutritional deficits secondary to dietary restriction or malabsorption (for example Vitamin B12, folate, Vitamin E or copper may cause an acquired form of progressive myelopathy, potentially reversible through nutritional supplementation [70]). Among metabolic diseases, biotinidase deficiency (an inborn error of biotin metabolism) may present with a progressive or recurrent form of acquired myelopathy with or without other neurological or ophthalmic manifestations [73,74]. These conditions, although rare, are potentially treatable and therefore should be considered after the commonest forms of progressive myelopathy were excluded. Metabolic storage disorders such as mucopolysaccharidoses or mucolipidosis may cause progressive myelopathy by spinal cord compression, or by direct involvement of the spinal cord white matter, such as in X-linked adrenoleukodystrophy, metachromatic leukodystrophy or Krabbe disease [75]. Among the neurodegenerative genetic syndromes, hereditary spastic paraplegias (HSPs) are a clinically and genetically heterogeneous group of disorders characterised by progressive weakness and lower limb spasticity [76]. Key characteristics of these myelopathies are summarised in Table 2C.

### 3.6. Functional Disorders

Functional disorders should also be considered when evaluating a child with symptoms suggestive of acute myelopathy, often as a diagnosis of exclusion. Functional disorders may present with motor symptoms including complete paralysis, gait disturbance, incoordination, tremors, or sensory symptoms such as pain or paraesthesia. Children may also complain of visual impairment, fatigue or headaches. Identifying incongruities in neurologic examination (such as non-anatomical sensory loss, motor inconsistency, collapsing weakness) and evidence of symptoms fluctuating, especially when the child is distracted or alone, may be helpful in diagnosis [77,78]. Features are summarised in Table 2C.

## 4. Diagnosis and Management

Early recognition of myelopathy is key, with detailed history and examination to narrow the broad differential diagnosis and direct further investigations. All children presenting with symptoms of a spinal cord syndrome (as detailed in Table 1) require urgent neuroimaging. For a subsection of patients in whom compressive myelopathy is suspected, for example, those with a history of trauma, a background of malignancy, or more rarely, those with disorders such as the mucolipidoses with bony involvement, neuroimaging should be very urgent (under 24 hours) preferably at a neurosurgical centre to facilitate urgent surgical management [80].

If myelitis is confirmed on imaging, further investigations may then be required to determine etiology. These are detailed in the algorithm described in Figure 3, alongside next steps in the case of a patient with clinical signs and symptoms of myelopathy where imaging does not confirm myelitis. Alongside MRI of both the brain and spinal cord with gadolinium contrast, other important investigations may include lumbar puncture (looking for CSF cell counts and differential, protein level, oligoclonal bands, and infectious causes such as bacterial/viral culture and PCR) and serum auto-antibodies against aquaporin-4 and myelin oligodendrocyte glycoprotein (MOG). Other investigations should be directed by clinical features, for example if MS or NMOSD are suspected assessment for optic neuropathy with visual-evoked potential and optical coherence tomography may be of value.

There will be specific management considerations depending on the cause of the myelopathy identified, which may be medical or surgical in nature. Early treatment is of great importance in many cases, with delays in initiating treatments such as plasma exchange in NMOSD having significant impacts on recovery and residual disability [51]. Supportive management however is important for all patients, with a holistic approach to what may be a devastating and disabling condition for many young people and families. Figure 4 summarises several important areas to consider in the management of the child with a myelopathy. Acutely, emergency management may include airway support for those with lesions high in the cord, with some patients requiring intensive care admission and mechanical ventilation [65].

Early physiotherapy and occupational therapy are key to maintaining range of movement, and should be ongoing to maximise rehabilitation, with improvements in strength and function possible for months and years after presentation [81]. Physiotherapy may include exercise therapy, hydrotherapy, and the use of robotic devices that aim to improve gait [82]. A variety of orthotics and assistive devices are also available, which are individualised based on age, severity of impairment and the child and family’s preferences [83]. Different members of the multidisciplinary team may need to be involved in developing comprehensive rehabilitation programmes, for example bowel and bladder retraining, planning adaptations to facilitate the return home and to school, and psychological support for children and families [82].

Going forward, there are several emerging treatments for myelopathies. One of these is nerve transfer, where segments of functioning “donor” nerves that innervate muscles considered expendable are diverted to supply muscles below the level of the spinal cord lesion. This was, so far, utilised to good effect in acute flaccid myelitis patients with poor clinical recovery [84] and patients with traumatic spinal cord injuries [85].

Spinal cord stimulation is another approach that uses electrical currents to assist rehabilitation of movement. A variety of strategies exist, from short-term transcutaneous stimulation used early on in rehabilitation to aid recovery, to more complex neuroprostheses that can be used to facilitate specific movements [86].

## Figures and Tables

**Figure 1 children-08-01055-f001:**
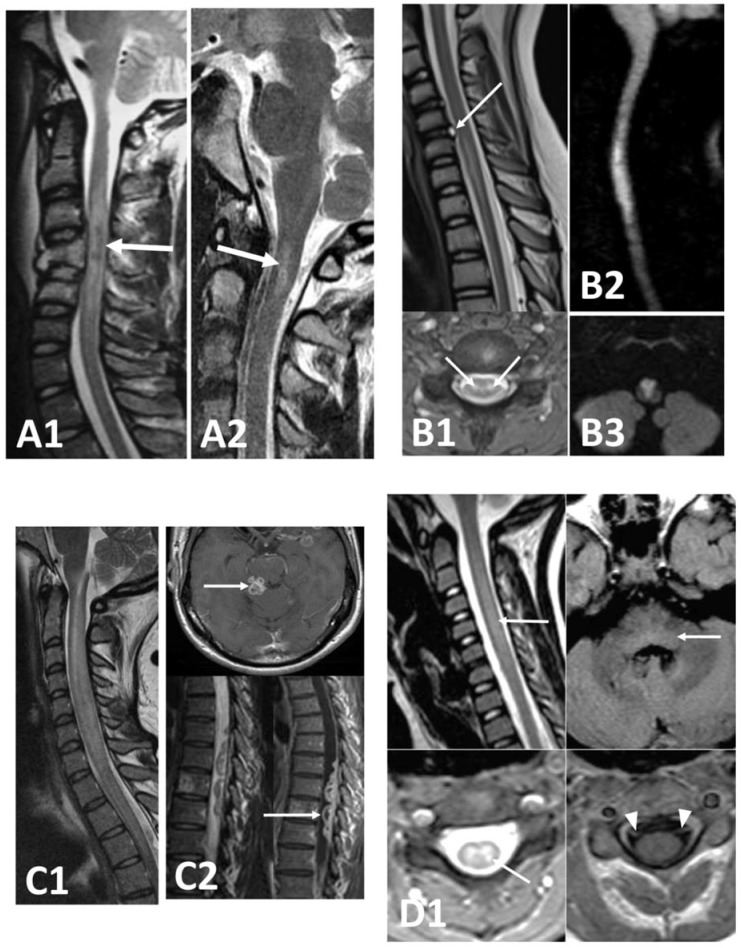
MRI imaging in paediatric myelopathies. (**A**) Traumatic myelopathies. Two patients. (**A1**)—blunt trauma in a young patient with a high velocity road traffic accident showing burst fractures, canal compromise and cord contusions with haemorrhage (arrow) and oedema. (**A2**)—penetrating trauma in a teenager stabbed in back of neck with normal bony spine on CT but cord contusion and myelopathy at C1-2 level on MRI (arrow). (**B**) Cord Infarction. (**B1**)—sagittal and axial images in a patient with sudden arm and leg weakness and preceding neck pain, showing an acute cervical disc prolapse (long arrow) and signal change in cervical cord localising to central grey matter (short arrows). (**B2**, **B3**)—two different patients with cord infarction showing utility of diffusion weighted imaging demonstrating restricted diffusion (high-signal) in cervical cord infarct and anterior cervicomedullary anterior spinal artery infarcts. (**C**) Infective Myelitis. (**C1**)—immunocompromised patient with myelitis and CMV in CSF with CMV viraemia showing a long segment cervical cord signal abnormality and swelling. (**C2**)—patient with proven TB meningitis presenting with bilateral leg weakness showing multiple conglomerate meningeal-based ring enhancing lesions in spinal canal (short arrows) and cranial cavity (long arrows) with extensive signal change in spinal cord. (**D**) Acute Flaccid Myelitis. (**D1**)—young patient with acute onset of bilateral limb weakness preceded by flu-like symptoms and positive enterovirus D68 on nasopharyngeal aspirate showing a long segment of cord signal change localising predominantly to central grey matter on axial images (arrows on sagittal and axial T2 images). Additional images of brainstem show dorsal pontine signal change in tegmentum (small arrow) and contrast enhancement of ventral cervical nerve roots (arrowheads).

**Figure 2 children-08-01055-f002:**
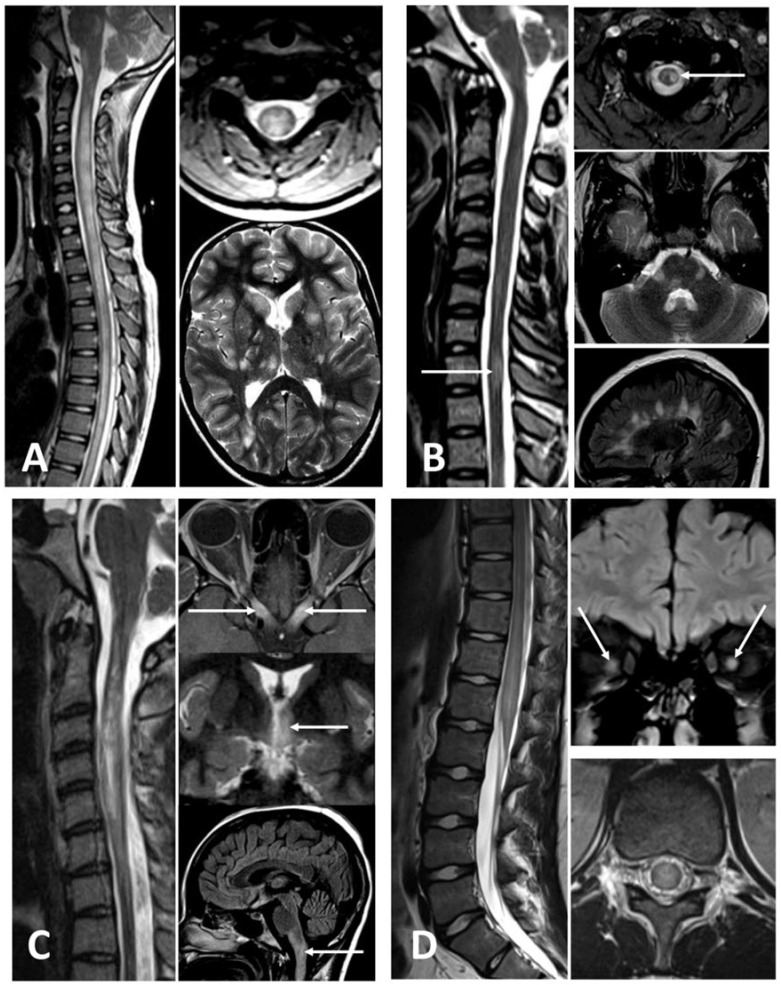
Paediatric myelopathies secondary to demyelinating diseases. (**A**) ADEM. Young patient presenting with encephalopathy and going ‘off legs’ showing extensive spinal cord signal change throughout, affecting both central and peripheral portions, with additional multifocal white matter, deep grey matter, and brainstem lesions typical for ADEM. (**B**) MS. Typical multiple, small, and short segment cord lesions (arrows) and additional similar peripheral brainstem lesions and further typical periventricular plaques in brain. (**C**) Neuromyelitis Optica. Young patient with a longitudinally extensive myelitis and some cavitary changes in central cord. Patient subsequently presented with an additional episode of bilateral posterior optic neuritis (long arrows), hypothalamic signal change (small arrows) and cervicomedullary involvement. CSF was positive for AQP4. (**D**) MOG related demyelination. Long lower thoracic cord lesion extending to conus, with both grey and white matter involvement on axial images and additional bilateral optic neuritis (arrows).

**Figure 3 children-08-01055-f003:**
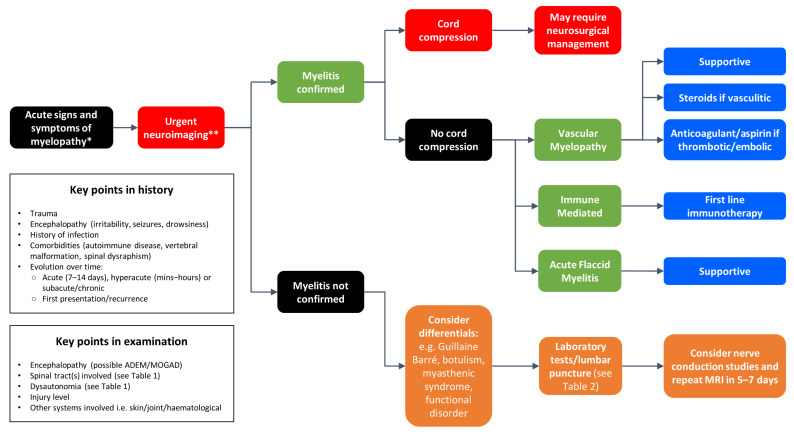
Approach to diagnosis and management of paediatric myelopathy. * See Table 1 for presenting features. ** If signs/symptoms of cord compression are present, imaging should be undertaken within 24 hours of presentation, preferably at a neurosurgical centre.

**Figure 4 children-08-01055-f004:**
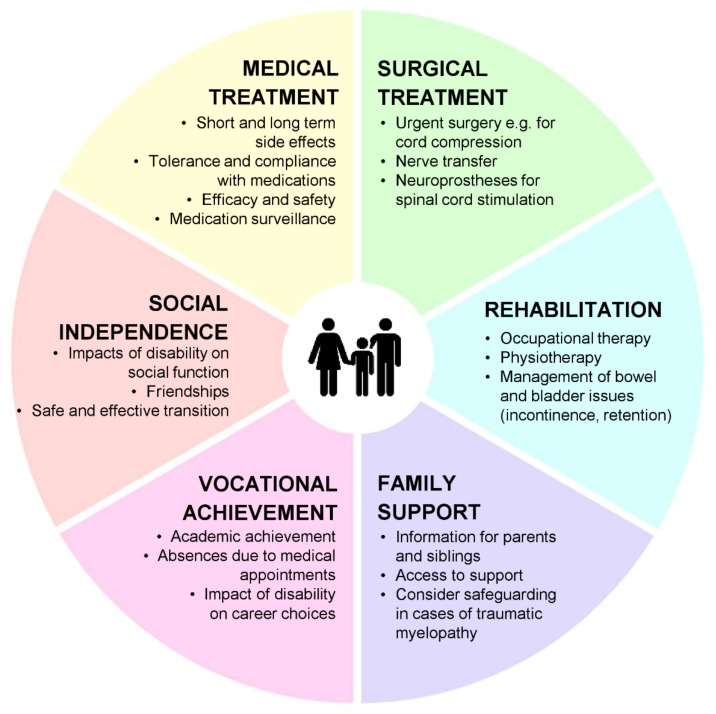
Holistic management of child with a myelopathy.

**Table 1 children-08-01055-t001:** Spinal cord syndromes in acute and chronic myelopathies [5,6,9,11,12,13].

Spinal Cord Syndrome	Possible Etiologies	Spinal Tracts Involved	Clinical Syndromes
Transverse cord syndrome	Acute transverse myelitis (demyelination, infectious, autoimmune)	All	Anaesthesia below the lesion Bilateral motor symptoms * Dysautonomia **
Lateral half cord syndrome (Brown-Sequard Syndrome)	Demyelination Extrinsic cord compression Trauma	- Ipsilateral dorsal column and corticospinal tracts - Contralateral spinothalamic tract	Ipsilateral loss of vibration, discriminative touch, and proprioception Ipsilateral motor symptoms* Contralateral loss of pain, nondiscriminative touch and temperature perception
Anterior cord syndrome	Anterior spinal artery occlusion Demyelination	- Spinothalamic tracts - Corticospinal tracts	Back pain Loss of pain, nondiscriminative touch and temperature perception Motor symptoms * +/− Dysautonomia **
Central cord syndrome	Syringomyelia Trauma (i.e., cervical hyperflexion or hyperextension) Demyelination	- Spinothalamic tracts - Corticospinal tracts	Loss of pain, nondiscriminative touch and temperature perception Motor symptoms* Upper extremities usually greater affected than lower +/− Dysautonomia **
Conus medullaris syndrome	Trauma Demyelination	Lesion at T12-L2 level	Bladder/bowel/sexual dysfunction prominent Bilateral leg weakness Saddle anaesthesia
Cauda equina syndrome	Trauma Spinal cord compression	Lesion at or below L3-L5 level Nerve roots	
Posterior cord syndrome	Posterior spinal artery occlusion Metabolic disorders (i.e., Vitamin B12 deficiency)	Dorsal columns	Loss of vibration, discriminative touch and proprioception Lhermitte’s sign
Selective anterior horn cell involvement	Viral infections (poliovirus, enteroviruses, flavivirus)	Secondary motor neurons	Bilateral flaccid paralysis, hyporeflexia, muscle atrophy, fasciculations

* Motor symptoms: acute phase; (spinal shock): flaccid paralysis, hyporeflexia. Chronic phase: pyramidal signs (spastic paralysis, hyperreflexia, upgoing plantars). If secondary motor neuron damage: mixed clinical picture, with persistent flaccid paralysis at the level of injury. ** Dysautonomia: dependent on level of injury, may include cardiovascular, bladder, bowel, and sexual dysfunction.

**Table 2 children-08-01055-t002:** **(A**) Characteristic features of acute nontraumatic myelopathies in childhood: infectious and vascular. (**B**) Characteristic features of acute nontraumatic myelopathies in childhood: inflammatory. (**C**) Characteristic features of acute nontraumatic myelopathies in childhood: other.

(A)	
Bacterial infections Abscesses (intramedullary, epidural) [53,56,57] Others: Borrelia burgdorferi (Lyme disease) [58] Bartonella henselae (Cat scratch disease) [59] Mycoplasma spp. [60]	Clinical picture of SIRS/Sepsis Spinal Dysraphism (i.e., dermal sinus) Clinical history of tick bite in endemic area for Lyme disease; useful diagnostic tests: Serum IgG and IgM antibodies CFS bacterial PCR
Acute viral infections [6,61,62,63,65,67,68,69,70] Herpesviruses: HSV, VZV, CMV, HHV6-7, EBV; Picornaviruses: Enterovirus-70, -71, -68, Coxsackieviruses A and B, Echoviruses, Polioviruses, Hepatitis A virus; Flaviviruses: WNV, Japanese encephalitis virus St. Louis encephalitis virus, Tick-borne encephalitis virus, Dengue virus; Influenza A virus, Measles virus, Mumps virus, Adenovirus	Clinical features of ATM or AFM Useful diagnostic tests: Nasopharyngeal aspirates/swab immunofluorescence assay for respiratory viruses Throat swab and stool for enterovirus PCR CFS Viruses PCR Serum IgM and IgG (acute and convalescent phase)
Chronic viral infections [71,72] HTLV-1 (Tropical Spastic Paresis) HIV (Vacuolar Myelopathy)	Chronic progressive myelopathy Useful diagnostic tests: CFS Viral PCR Serum IgM IgG
Vascular Myelopathies [30,34]: Hypotension (systemic) Intravascular injury (Iatrogenic, minor or major trauma) Cerebellar herniation Thrombotic disease Vasculitis Embolic disease Microcirculatory compromise (arteriovenous malformation) Atlanto-axial instability	Hyper-acute onset (minutes to hours); History of trauma, systemic hypotension, hypercoagulable state, vasculitis, surgery (iatrogenic) MRI: spinal cord swelling, hyperintense on T2 with a “pencil-like” shape
(**B**)	
Idiopathic Acute Transverse Myelitis [3]	Acute or subacute onset (2 hours–21 days) Bilateral motor and sensory impairment (may be asymmetric) MRI cord and brain: central cord T2 hyperintensities, +/− T1 hypointensities, +/− brain lesions Clear sensory level not always evident Exclusion of other causes of ATM
Connective tissue disorders [3,39] Systemic Lupus Erythematosus, Sjögren syndrome, antiphospholipid antibody syndrome, Behçet disease	Clinical picture of systemic inflammation/multisystemic involvement. Serum: ANA, ANCA, RF, anti–dsDNA, antiphospholipid antibodies, Angiotensin-converting enzyme level;
Multiple Sclerosis [44]	McDonald MRI MS criteria; CSF oligoclonal bands
Neuromyelitis Optica Spectrum Disorders [47]	Optic neuritis; LETM Serum: Aquaporin 4 IgG Acute brainstem syndrome, area postrema syndrome, diencephalic syndrome
Acute Disseminated Encephalomyelitis [45]	Encephalopathy; brain MRI: multiple hyperintense bilateral, asymmetric patchy and poorly marginated lesions
Myelin Oligodendrocyte Glycoprotein (MOG) Antibody-associated Disease [46]	Optic neuritis, may be bilateral LETM, may be multifocal Can present with ADEM or ADEM-like syndrome of encephalopathy Serum: positive MOG antibodies
**(C)**	
Functional disorders	Motor symptoms: paralysis, gait disturbance, incoordination, tremor, loss of speech. Sensory symptoms: paraesthesia, intractable pain, tunnel vision, and blindness. Others: headache, fatigue, and nonepileptic seizures Diagnostic clues: identification of incongruities; short remission when alone or in emergencies [78]
Nutritional deficiency:	
Vitamin B12/Folate	Posterior and lateral columns in the cervical and thoracic segments of the spinal cord. Symmetric loss of vibration and proprioception especially lower limbs; +/− spastic paraparesis; +/− bowel/bladder dysfunction; other signs and symptoms: peripheral neuropathy; psychiatric symptoms; megaloblastic anaemia [73].

Copper	Clinical presentation resembling Vitamin B12/folate deficiency; other signs and symptoms: peripheral axonal neuropathy, optic neuropathy [73].
Vitamin E	Posterior column degeneration and peripheral neuropathy: ataxic gait; reduced/absent deep tendon reflexes Other signs and symptoms: peripheral neuropathy, ophthalmoplegia, pigmentary retinopathy, myopathy [73].
Hereditary spastic paraplegias (HSPs) [76] Genetic forms: autosomal dominant, autosomal recessive, and X-linked recessive	Corticospinal tracts and posterior columns involvement (longest fibers): lower limb spasticity, pyramidal weakness hyperreflexia, extensor plantar responses. Early/Late onset forms Complicated form: presence of other system involvement (i.e., ataxia, epilepsy, muscle atrophy, optic atrophy, pigmentary retinopathy)
Metabolic storage diseases: [75,79]:	
Mucopolysaccharidoses/Mucolipidoses	Neuro-developmental regression, coarse facial features, dysostosis multiplex, organomegaly, ocular manifestation, multisystemic involvement.

X-linked adrenoleukodystrophy	Adrenomyeloneuropathy phenotype: long tract degeneration in the spinal cord, late adolescence to adult onset with progressive paraparesis.

Metachromatic leukodystrophy	Upper and lower motor neuron lesions, cognitive and psychiatric symptoms. In the infantile form irritability, gait disturbance, knee hyperextension.

Krabbe disease	Spasticity and normal or absent deep tendon reflexes Irritability, cognitive impairment, seizures

## Data Availability

Not applicable.

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
