# Peer review of "Acute Myelopathy in Childhood"

_children, 2021, doi:10.3390/children8111055_

Round 1
Reviewer 1 Report
This is a nicely written review. I recommend to publish it as written.
Author Response
Many thanks for your review of our manuscript and your positive comment.
Reviewer 2 Report
I salute the authors for the comprehensive review that provide a good insight on the pathology and its management.
Author Response

(The authors gave the same response as above.)

Reviewer 3 Report
The present paper in a fine review of acute myelopathy in childhood. Quite a few reviews on this topic can be found in the literature, but only a few of them are focused on the paediatric age.
No ethical concerns on my part.
The contents and the writing of the article are clear, interesting and useful for the reader. Figures and table adequately support the text.
There is only one aspect that I believe needs to be reviewed and that is the rehabilitation intervention that should be carried out as early as possible to promote the highest degree of recovery and minimize the sequelae. It is, in my opinion, necessary to rewrite the paragraph between lines 395 and 400, given that the sequelae of acute myelopathies leave a high percentage of cases with a high degree of disability, I believe it is necessary to mention the comprehensive rehabilitation programmes for spinal cord injury, which include physical treatment (respiratory physiotherapy, kinesitherapy, occupational therapy, bladder and bowel re-education, prescription of orthoses and technical aids such as wheelchairs or standing frames, mechanotherapy and robotic treatment of the upper limb and gait, intervention of the child and his or her family and the planning of the necessary adaptations for the return to home and school. For a comprehensive and adequate overview of the required fragment, see the following quote:
1. Vogel LC, Betz RR, Mulcahey MJ. Spinal cord injuries in children and adolescents. Handb Clin Neurol 2012; 109:131-48.
2. Parent S, Mac-Thiong JM, Roy-Beaudry M, Sosa JF, Labelle H. Spinal Cord Injury in the Pediatric Population: A Systematic Review of the Literature. J Neurotrauma 2011; 28: 1515–24.
3. Nikolay Peev , Alexander Komarov , Enrique Osorio-Fonseca , Mehmet Zileli. Rehabilitation of Spinal Cord Injury: WFNS Spine Committee Recommendations. Neurospine 2020;17(4):820-832.
Author Response
Thank you for your comments and for highlighting this very important point. We have rewritten the paragraph between lines 395 and 400 (now lines 396-405) to further elaborate on the rehabilitation of these patients, using two of the papers (Vogel et al., 2012 and Peev et al., 2020) suggested.